# A Causality Preserving Evolution of a Pair of Strings †

**Serguei Krasnikov**

Central Astronomical Observatory at Pulkovo, 196140 St. Petersburg, Russia; s.v.krasnikov@mail.ru
† To Milena.

**Abstract:** As follows from Gott's discovery, a pair of straight string-like singularities moving in opposite directions, when they have suitable speed and impact parameter, produce closed timelike curves. I argue in this paper that there always is a not-so-frightening alternative: the Universe may prefer to produce a certain (surprisingly simple and absolutely mild) singularity instead.

**Keywords:** causality; gott pair; flat spacetime

## 1. Introduction

### 1.1. The Objective

The subject matter of this paper is some properties of a very special type of singularity. Suppose, a spacetime admits a finite open convex covering

$$M = \bigcup_{i=1,\ldots i_0 < \infty} M_i \tag{1}$$

such that each $M_i$ is a subset of a Minkowski $n$-dimensional space $\mathbb{L}^n$. Such spacetimes are exceptionally simple and this enables one [1] to assign a "form" to singularities contained there (throughout the paper, we stick to the physical level of rigor and drop discussion of "self-evident" concepts and facts). When such a singularity has the form of an $(n-2)$-dimensional surface, the former is called a string-like singularity or just a string.

String-like singularities are abundant in GR, see [2–4] for some reviews and references, and [1] for singularities (1) of less trivial forms. For instance, in the ($n = 3$) case, the singularities are associated with world lines of massive pointlike particles and in the ($n = 4$) case their "surface-like" counterparts–approximately–describe cosmic strings (for an example of a use of this approximation, see [5]).

Still, string-like singularities are not to be confused with cosmic strings. They are objects of different nature. The former, in particular, are purely geometric entities. In contrast to the latter, they are everywhere flat, which implies, in particular, that they solve the vacuum Einstein equations. Moreover, they do not bend or, say, emit gravitational waves.

Not much is known about these strings' possible dynamics. Until recently, the only relevant result was that obtained by Hellaby [6] who proved that mutually perpendicular string-like singularities do not pass intact through each other. Another, almost concurrent, result was due to Gott [5]. Loosely speaking, it says that two parallel strings in $\mathbb{L}^4$ or, equivalently, 2D cones in $\mathbb{L}^3$ having the angle deficit $\alpha$ and moving in opposite directions with some particular speed $v < c$, and impact parameter $d$ (a Gott pair), produce closed timelike curves. True, it was claimed in [7] that the initial conditions at spacelike infinity of Gott's spacetime are unphysical, but as pointed out by Headrick and Gott [8], unphysicality in [7] is postulated rather than derived. The goal of this paper is to prove by construction that the causality violation is unnecessary: for any $d$, $\alpha$, and $v$ there is an inextendible spacetime, $S_f$, which describes the said scattering, but in which the causality condition holds everywhere.

*1.2. Gluing Spacetimes*

All relevant spacetimes below are built from Minkowski space by a certain manipulation called "gluing" and in this subsection, we provide a tolerably formal meaning to that word in application to portions of spacetimes.

From a spacetime $M$ remove a subset $U$ such that the space $N \equiv M - U$ is a spacetime with boundary. Now suppose that for some components of the boundary, $\partial N_1, \partial N_2$ there is a neighbourhood $O \supset \partial N_1$ and an embedding $\sigma \colon O \to M$ such that

$$O \cap \sigma(O) = \varnothing, \qquad \sigma(O - N) \subset N, \qquad \sigma(\partial N_1) = \partial N_2. \tag{2}$$

Then

$$S \equiv N \cup_\sigma O \tag{3}$$

is said to be the result of gluing together $\partial N_1$ and $\partial N_2$, cf. Figure 1, by $\sigma$.

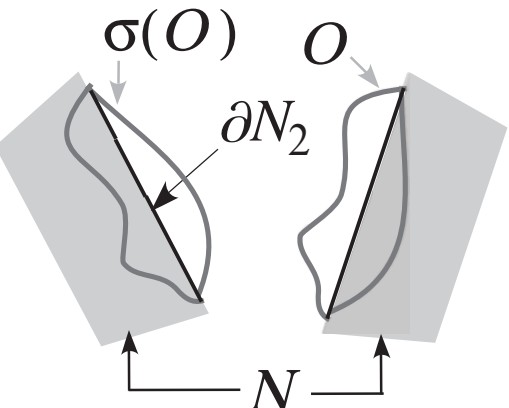

**Figure 1.** The white area is $U$.

## 2. Elementary Examples

Presently, we are going to consider a couple of simple specific examples. They are interesting by themselves and also they will be useful in the next section.

**Example 1** (Straight string)**.** *Let $U$ be the dihedral angle swept in the course of evolution by a resting 2D cone $\mathcal{A}$*

$$U = \bigcup_t \mathcal{A}_t, \quad \mathcal{A}_{t_0} \equiv \{p \colon \quad t(p) = t_0, \quad -\alpha^*/2 < \phi(p) < \alpha^*/2\}. \tag{4}$$

*in Minkowski space*

$$\mathrm{d}s^2 = -\mathrm{d}t^2 + \mathrm{d}\,r^2 + r^2 \mathrm{d}\phi^2, \qquad t \in \mathbb{R}, \quad r > 0, \quad \phi \text{ is identified with } \phi + 2\pi \tag{5}$$

*where $\alpha^*$ is a non-zero constant smaller than $\pi$.*

*Then, the components of the boundary are the half planes $\partial N_{1,2} = \{p \colon \phi(p) = \mp\alpha^*/2\}$, the neighbourhood $O$ can be chosen to be the wedge $\{p \colon -\alpha^*/2 - \epsilon < \phi(p) < -\alpha^*/2 + \epsilon\}$ and $\sigma$ be the rotation by $\alpha^*$ in the planes of fixed $t$. It is the thus obtained spacetime $S$—referred to as $S_A$ in this particular case—that is usually called string.*

**Example 2** (Running angle. Figure 2)**.** *In order to construct a mild generalization (depicted in Figure 2) of the string singularity discussed above, let us redefine $U$*

$$U \equiv \bigcup_t \mathcal{B}_t, \qquad \mathcal{B}_{t_0} \equiv \tau_{t_0}(\mathcal{A}_{t_0}), \tag{6}$$

*where $\tau(\cdot)_{t_0}$ is the translation by $vt_0$ in the x-direction (x being a cartesian coordinate $x \equiv r \cos \phi$) with $v \equiv const < c$ and $v \sim \vec{Ox}$. Obviously, the just built spacetime, $S_B$, and the former one, $S_A$, are* isometric, *they are related by a boost in the x-direction.*

　　　*Two circumstances are especially noteworthy:*

1.　*pick a point p of the upper gray ray in Figure 2. The deficit angle of the moving string, $\alpha$, is*

$$\alpha = 2 \arctan[y(p)/x(p)]$$

　　*whence*

$$\alpha^* = 2 \arctan[y^*(p)/x^*(p)] = 2 \arctan\{y(p)/[\gamma(v)x(p)]\}.$$

　　*The asterisk here denotes "in the proper reference system of the string" and $\gamma$ is the Lorentz factor. Thus*

$$tg\frac{\alpha}{2} = \frac{1}{\sqrt{1-v^2}} tg\frac{\alpha^*}{2}; \tag{7}$$

2.　*the rotation axis of the moving string $S_B$ is not parallel to the t-axis. Correspondingly, a vector initially lying in the $(x,y)$-plane acquires, after being transported around the string, a nonzero t-component. This means, in particular, that t is a "bad" coordinate: the (maximal extensions of) surfaces $t = const$ are not embedded into the spacetime.*

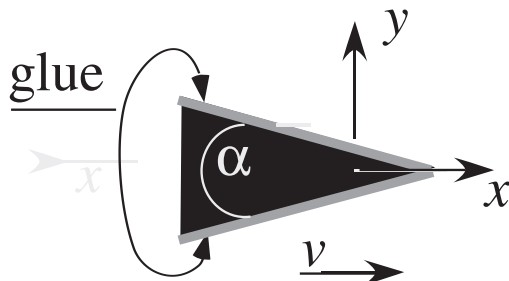

**Figure 2.** The black angle is the "hole" $\mathcal{B}_{t_0}$ for a certain $t_0$ (to convert the former to $\mathcal{A}_{t_0}$ set $v = 0$, $\alpha = \alpha^*$). $S_B$ is obtained by identifying—at each $t$—the upper gray ray, $y = -tg(\alpha/2)(x - vt)$, $y > 0$, with the lower one, $y = tg(\alpha/2)(x - vt)$, $y < 0$.

**Example 3** (Inelastic head-on collision). *Presently, consider the spacetime with*

$$U = U_C \equiv \bigcup_t \mathcal{C}_t, \qquad \mathcal{C}_{t_0} \equiv \mathcal{B}_{-t_0} \cup \rho_Y(\mathcal{B}_{-t_0}), \tag{8}$$

*where $\mathcal{B}$ are defined in (6) and $\rho_Y(\cdot)$ is the reflection through the y-axis in the $(x,y)$-plane. U at each moment of t is a pair of equal angles moving—with the speed v—towards each other until at $t = 0$ their vertices collide, see Figure 3. At positive t, $\mathcal{B}_{-t}$ and $\rho_Y(\mathcal{B}_{-t})$ start to overlap, see Figure 3c. Or, they can be viewed as a pair of receding obtuse angles (bounded by the gray lines in Figure 3) either of which has magnitude $\pi - \alpha$ and (vertically directed) velocity $vtg\frac{\alpha}{2}$. The spacetime S, denoted $S_C$ in this case, describing the head-on collision of two cones (or two parallel strings) is obtained by the pairwise gluing together—at each t—the upper two gray rays $y = tg(\alpha/2)(|x| + vt)$, $y > 0$ with the lower two $y = -tg(\alpha/2)(|x| + vt)$, $y < 0$.*

　　*$S_C$ contains no closed causal curves. This observation is not quite trivial, as is observed from the comparison between $S_C$ and a Gott pair. The proof is based on the fact that $\sigma$ obeys the condition*

$$t(\sigma(p)) = t(p), \qquad \forall p \in \partial N_i \tag{9}$$

*(i.e., only points with the same t are identified). t grows along any future directed causal curve and hence such a curve cannot be closed.*

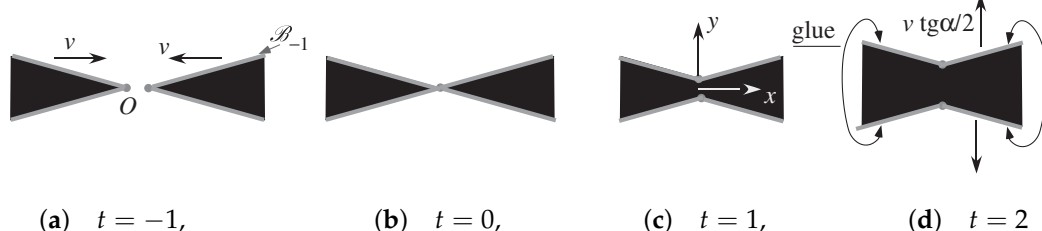

**(a)** $t = -1$,         **(b)** $t = 0$,         **(c)** $t = 1$,         **(d)** $t = 2$

**Figure 3.** The cross sections $t = const$ of $N$ (the white area) and of $\partial N$ (the gray rays). $S_C$ is a pair of strings merging into a single resting one.

### 3. String—String Scattering

In this section, we finally present a spacetime that can be interpreted as a causality respecting evolution of a Gott pair.

We start with the spacetime $U_X \subset \mathbb{L}^3$, which differs from $U_C$, see Equation (6), by one detail: the translation $\tau$ is changed to the superposition $\delta \circ \tau$, where $\delta$ is the translation by $d/2$ in the $y$-direction. The cross-section $t = t_0 < 0$ of $U_X$ is a pair of angles, moving towards each other with speed $v$ and with non-zero impact parameter $d$, see Figure 4a.

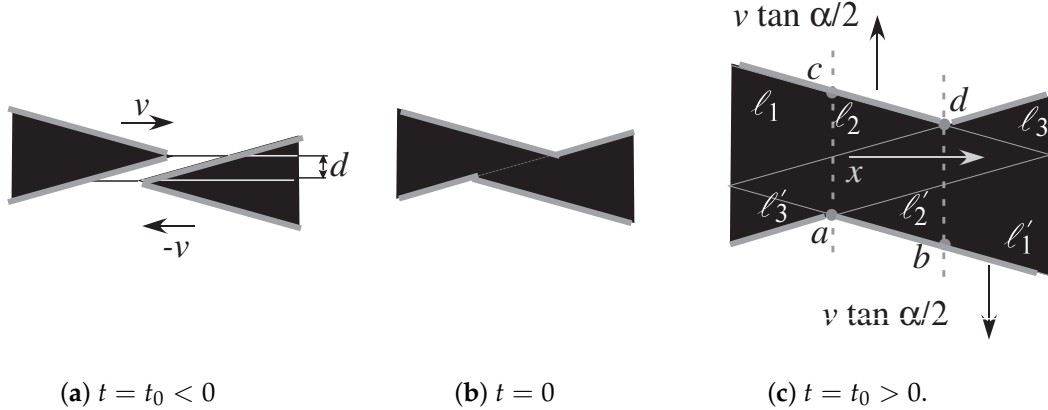

**(a)** $t = t_0 < 0$         **(b)** $t = 0$         **(c)** $t = t_0 > 0$.

**Figure 4.** The gray lines are the sides of corresponding angles.

At positive $t$'s, the angles partially overlap, taking the form of a skewed bowtie, see Figure 4c. The bowtie's boundary is a pair of broken lines, related by the point reflection $\omega$ through the origin, of which the upper one consists of the straight segments $\ell_1 \equiv (-\infty, c)$, $\ell_2 \equiv (c, d)$, and $\ell_3 \equiv (d, \infty)$. Correspondingly, the lower broken line is constituted by the segments $\ell_i' \equiv \omega(\ell_i)$, $i = 1, \ldots, 3$. In the course of evolution, all four vertices $a, b, c, d$ change their location and each $\ell_i^{(')}$ sweeps a strip $\mathcal{L}_i^{(')}$.

Presently, glue $\mathcal{L}_3'$ to $\mathcal{L}_1$ and $\mathcal{L}_3$ to $\mathcal{L}_1'$ (the gluing isometries being the rotation with the duly tilted axes, cf. Example 2). The resulting spacetime, $R$, has almost all properties, cf. Conclusions, of the sought-for spacetime $S_f$. The former, however, is extendible (that is there exists a spacetime $X$ "greater" than $R$, i.e., $R \subsetneq X$). To eliminate this last "flaw", let us, first, introduce one more object—the parallelogram $(p', p, q, q')$ depicted by the white quadrangle in Figure 5 and defined as the parallelogram $(a, c, d, b)$, see Figure 4c, contracted in the vertical direction so that

$$\begin{aligned} \forall t \leqslant 0 \quad & (p', p, q, q') = \varnothing, \\ \forall t > 0 \quad & y(c) - y(p) = y(p') - y(a) = const. \end{aligned} \tag{10}$$

Of course the locations of the points $p^{(')}$, $q^{(')}$ are again functions of $t$. Thus, with the passage of time, the segments

$$(p'q'), \quad (p'p), \quad (qq'), \quad (pq)$$

sweep four strips, which we shall denote by $\mathcal{K}$ with corresponding indexes.

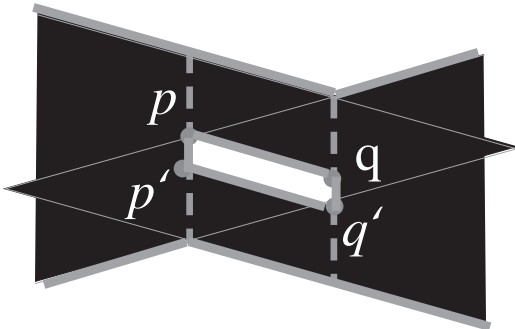

**Figure 5.** The cross section of the middle part of $S_f$. At each positive $t$ the following segments are identified: $(p'q') = l'_2$, $(pq) = l_2$, $(p'p) = (q'q)$.

The last step in building $S_f$ is gluing together

$$\mathcal{K}_{pq} \text{ and } \mathcal{L}_2, \quad \mathcal{K}_{p'q'} \text{ and } \mathcal{L}'_2, \quad \mathcal{K}_{pp'} \text{ and } K_{qq'}$$

which is possible because, as follows directly from the definition, $\mathcal{K}_{pq}$, $\mathcal{K}_{p'q'}$, and $\mathcal{K}_{pp'}$ have the same proper widths as and are parallel to $\mathcal{L}_2$, $\mathcal{L}'_2$, and $\mathcal{K}_{qq'}$, respectively (note that though $\ell_2(t)$ and $\ell'_2(t)$ are parallel, the map sending one of them to the other fails to do the same with their velocities. That is why we cannot use the translation as a gluing isometry between them).

Thus, one can obtain the desired spacetime $S_f$ by gluing together certain surfaces. In doing this, one identifies only points with the sane $t$. Therefore, by the criterium (9), $S_f$ contains no closed causal curves.

## 4. Conclusions

In summary, we have demonstrated that for any $d$, $\alpha < \pi$, and $v < c$ there is a *causality respecting* spacetime, $S_f$, which describes the scattering with the impact parameter $d$ of two strings. Either moves in an "otherwise Minkowski" space with the speed $v$ and has the angle deficit $\alpha$.

From the Minkowski space $M$

$$\mathrm{d}s^2 = -\mathrm{d}t^2 + \mathrm{d}\,x^2 + \mathrm{d}y^2, \qquad t, x, y \in \mathbb{R}.$$

remove the wedge

$$W \equiv \{p \in M: \quad t(p) > k|x(p)|\}, \quad k = const > 1$$

and glue together the boundaries

$$\partial N_{1,2} \equiv \{p \in M: \quad t(p) = k|x(p)|, \ x(p) \lessgtr 0\}.$$

The resulting spacetime, $T_k$, called a *tachyonic string* is similar to that considered in Example 1 and describes a superluminal particle.

It is readily observed that at negative $t$, $M$ is isometric to $T_k$, but the whole spacetimes differ. Put another way, the $(t < 0)$-region of a Minkowski space has infinitely many different (varying in $k$) flat extensions. In fact, it is easy to prove (for example, by employing the notion of "loop singularity" [1]) a stronger fact:

**Proposition 1.** *Any spacetime M, if it has a flat extension $\tilde{M} \neq \mathrm{Cl}\,M$, has infinitely many different flat extensions, each of which contains a (loop) string-like singularity.*

Thus, in a theory, where string-like singularities are included, the uniqueness of evolution of a spacetime is out of the question [the opposite claim made in [9] should be taken with caution: in all appearance the author implies that of all imaginable singularities only those considered in Examples 1 and 2 (and their intersections) are allowed in spacetimes under study].

Thus, the existence of $S_f$ is by itself not surprising in the least. Moreover, it is a direct consequence of the theorem proven in [10,11]. What *is* surprising is that $S_f$ turns out to be so simple. In particular, it is orientable and string-like, cf. Equation (1).

**Funding:** This research received no external funding.

**Data Availability Statement:** Not applicable.

**Conflicts of Interest:** The author declares no conflict of interest.

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
