# Peer review of "A Causality Preserving Evolution of a Pair of Stringsâ€"

_universe, doi:10.3390/universe8120640_

Round 1
Reviewer 1 Report
The author studies the possible fate of a so-called Gott pair of cosmic strings. Historicalled this has garnered quit a bit of interest, because such pairs could possibly lead to the creation of closed timelike curves. Unfortunately, the auhtor forgoes any discussion of the historical work on this subject and previous conclusions. This is unfortunate, because it might have prevented the duplication.
The approach taken by the author of arranging the deficit angles of moving conical singularities in such a way that only points with the same time coordinate are ever identified, is highly reminescent of the approach taken by 't Hooft in https://doi.org/10.1088/0264-9381/9/5/015. Given that the method of 't Hooft should produce the unique future evolution of this system, the solution provided by the authors is either fully equivalent with that of 't Hooft, or shows a flaw in 't Hooft's argument.
In the first case, I don't see this becoming a publishable result. (Maybe unless a discussion is included on how this alternate formulation of the solution leads to new insights on the fate of a Gott pair.) In the second unlikely case, a clear discussion of how 't Hooft's method fails is required.
In either case, a major revision of the work to include a disccusion of its relationship to previous work is needed before it can properly be considered for publication.
Author Response
The reviewer requires a clear discussion of how ’t Hooft’s method fails.
Correspondingly, I have added a passage to the section ”Conclusions” explaining
that ’t Hooft achieves the uniqueness of his solutions by excluding—by fiat—
some singularities.
Reviewer 2 Report
In this article, it is proposed that the space-time corresponding to straight, infinitely-long and thin cosmic strings moving in opposite directions may not be the one derived by Gott in 1991, which has closed time-like curves, but instead a different one an additional singularity but no closed time-like curves. This space-time is constructed by “gluing” parts of Minkowski-like spaces together in a way that has no closed time-like curve.
While this paper is interesting, at the moment I fail to fully understand its mathematical or physical value, for the two following reasons
* Figures 2 and 4 (and the construction they illustrate) look unclear to me. Where are the two strings for t > 0? The drawings seem to imply that they are inside the deficit angles, which is of course impossible as this deficit angle is just a representation, not part of the actual space-time.
* Notice that the solutions with closed timelike curves are probably not physical anyway (see for instance https://journals.aps.org/prl/abstract/10.1103/PhysRevLett.68.267 and https://journals.aps.org/prd/abstract/10.1103/PhysRevD.72.043532). In practice, their mathematical existence thus does not seem to be a problem as, in any physically realistic scenario, they will not be generated in the first place.
I also have a few less important comments and suggestions:
* In the next-to last paragraph of section 1.1, the phase “Not much is known about these strings possible dynamics” is misleading: there is an extensive literature about the different types of string-like solutions of the Einstein equations, both with and without charged matter fields, their stability, relaxation from excited states, energy loss via loop formation, collisions of strings in different configurations (some of which propose a different resolution to the problem of closed time-like loops), and evolution of a gas of cosmic strings in an expanding universe, both numerical and analytical. I would suggest the author to cite at least some of these papers, especially those that are most relevant for this study.
* In the second paragraph of section 1.2, what are the conditions on M and U for this construction to work?
* In (5), the equation $\phi = \phi + 2 \pi$ looks a bit misleading. I would suggest replacing it by the statement than $\phi + 2 \pi$ is identified with $\phi$.
* I would suggest reminding the definition of an extendible space-time and explaining why adding the parallelogram (p q q' p') makes it non-extendable.
In conclusion, I am not sure I really understand why the construction makes sense either from a physical or a mathematical points of view. For this reason, I can not recommend publication of this paper in its current state. I would of course be happy to revise my opinion if the author can convincingly explain why this construction is mathematically sensible and physically sound.
Round 2
Reviewer 2 Report
I thank the author for his reply and changes to the manuscript which resolve some of my earlier concerns. It is, however, sill unclear to me what the set-up under study precisely is. More specifically, the two points I am unclear about are:
* The abstract explicitely mentions “a pair of straight cosmic strings”, while the body of the paper pertains to string-like singularities which “are not to be confused with cosmic strings”. I think that, in order to be suitable for publication, this paper should clarify whether the singularities under considerations are cosmic strings (or maybe a specific type thereof) or not.
* If the singularities under considerations are not cosmic strings, are they global solutions of the EInstein field equations? If yes, what could be the origin of the Dirac-like energy-momentum tensor at the tip of the conical deficits for t < 0, and is there a motivation for how its evolution can remain compatible with the configurations shown in Figure 2 and 4 at t > 0? (If I understand correctly, the configurations described in the article are by construction vacuum solutions of the Einstein field equations, and thus physically relevent, everywhere except on the singularities; but I am not sure I see why they remain relevant at t > 0 as it seems to me the energy-mometum tensor should change in a very specific manner for which I can't see a justification.)
For these reasons, I think this article is not suitable for publication in its current state, although I will be happy to revise my opinion if the two points above are answered.
